# Terrain Ruggedness and Canopy Height Predict Short-Range Dispersal in the Critically Endangered Black-and-White Ruffed Lemur

**DOI:** 10.3390/genes14030746

**Published:** 2023-03-18

**Authors:** Amanda N. Mancini, Aparna Chandrashekar, Jean Pierre Lahitsara, Daisy Gold Ogbeta, Jeanne Arline Rajaonarivelo, Ndimbintsoa Rojoarinjaka Ranaivorazo, Joseane Rasoazanakolona, Mayar Safwat, Justin Solo, Jean Guy Razafindraibe, Georges Razafindrakoto, Andrea L. Baden

**Affiliations:** 1Department of Anthropology, The Graduate Center, City University of New York, New York, NY 10016, USA; 2The New York Consortium in Evolutionary Primatology (NYCEP), New York, NY 10065, USA; 3Centre ValBio Research Center, Ranomafana, Ifanadiana 312, Madagascarjeanguyrazafindraibe@gmail.com (J.G.R.);; 4Department of Nursing, Helene Fuld College of Nursing, New York, NY 10035, USA; 5Department of Chemistry, Hunter College, New York, NY 10065, USA; 6UMI 233 TransVIHMI, Institut de Recherche pour le Développement (IRD), University of Montpellier, Inserm U 1175, 34000 Montpellier, France; 7Department of Zoology and Animal Biodiversity, Faculty of Science, University of Antananarivo, Antananarivo 101, Madagascar; 8Department of Anthropology, Hunter College, New York, NY 10065, USA

**Keywords:** Madagascar, isolation-by-resistance, circuit theory, gravity models, landscape genetics, conservation

## Abstract

Dispersal is a fundamental aspect of primates’ lives and influences both population and community structuring, as well as species evolution. Primates disperse within an environmental context, where both local and intervening environmental factors affect all phases of dispersal. To date, research has primarily focused on how the intervening landscape influences primate dispersal, with few assessing the effects of local habitat characteristics. Here, we use a landscape genetics approach to examine between- and within-site environmental drivers of short-range black-and-white ruffed lemur (*Varecia variegata*) dispersal in the Ranomafana region of southeastern Madagascar. We identified the most influential drivers of short-range ruffed lemur dispersal as being between-site terrain ruggedness and canopy height, more so than any within-site habitat characteristic evaluated. Our results suggest that ruffed lemurs disperse through the least rugged terrain that enables them to remain within their preferred tall-canopied forest habitat. Furthermore, we noted a scale-dependent environmental effect when comparing our results to earlier landscape characteristics identified as driving long-range ruffed lemur dispersal. We found that forest structure drives short-range dispersal events, whereas forest presence facilitates long-range dispersal and multigenerational gene flow. Together, our findings highlight the importance of retaining high-quality forests and forest continuity to facilitate dispersal and maintain functional connectivity in ruffed lemurs.

## 1. Introduction

Animal movement, or an individual’s change in spatial location through time, can occur at multiple spatial and temporal scales [1,2]. It is a fundamental aspect of a primate’s life and is key to individual survival and fitness, population and community structuring, and species evolution [1,3]. Behaviors involving close-range movement, such as foraging, locating a mate, or avoiding predators, occur periodically at relatively small spatial and temporal scales [1]. By contrast, long-range movement, such as natal or secondary dispersal, typically occurs only once or a few times throughout a primate’s lifetime and at relatively larger spatial and temporal scales. The latter also facilitates population connectivity through gene flow; has important consequences for adaptation and speciation [4,5,6]; and is one of the primary drivers of community structure and assembly across primates [3,7].

Primate movement occurs within an environmental context, and thus environmental variables play an explicit role in the motivation, capacity, timing, and direction of an individual’s movement decisions [1,8]. Large-scale environmental variability, particularly landscape composition, configuration, and quality, can significantly impact long-range movement decisions, including the timing and success of dispersal events [9,10]. For instance, many taxa preferentially disperse through landscapes that are most structurally similar to their source habitat [11,12]. In cases where landscapes are of poor quality or fragmented, dispersal can be delayed, dispersal rates decrease, and mortality risks increase, especially as distances between patches grow [13,14,15,16,17,18]. The quality of the source habitat can also influence an individual’s dispersal propensity and motivations [19,20]. Individuals inhabiting lower quality source habitats often exhibit higher, albeit delayed, dispersal rates and greater dispersal distances than those from higher quality sites [21,22,23,24,25,26]; but see [14].

Research directly investigating large-scale primate movement has historically been limited; individual dispersal events are rare and difficult to observe, as dispersing animals often settle large distances from their departure site [27,28]. Large-scale movements such as these can be monitored opportunistically with remote GPS tracking (e.g., wolves, *Canis lupus*: [29]; leopards, *Panthera pardus*: [30]; tigers, *Panthera tigris*: [31]), though it has only rarely been used in primates (e.g., *Papio ursinus*: [32]; *Daubentonia madagascariensis*: Louis pers. comm.). Instead, most studies of primate dispersal rely on indirect measures, among them genetic distance, which can be generated from population genetic and genomic data to quantify gene flow between groups or populations and infer functional population connectivity (i.e., successful dispersal followed by subsequent mating and reproduction; [33,34]). More recently, landscape genetic methods have allowed researchers to combine genetic distances with remotely sensed landscape information to explicitly evaluate the environmental drivers of between-site functional connectivity and within-site environmental variability and gene flow [35,36,37]. Studies of primate landscape genetics remain limited, though are increasing in number [38]. From these, it is clear that primate gene flow can be impeded by both natural (e.g., rivers: [39,40]) and anthropogenic barriers (e.g., highways: [41]), including anthropogenically-driven landcover change (e.g., agriculture and deforestation: [41,42,43,44,45]) and proximity to human settlements ([46,47]). Environmental variability can influence dispersal at multiple scales—from smaller-scale gene flow resulting from typical dispersal events to long-range dispersal and multigenerational gene flow—with implications ranging from driving local population genetic structure to influencing potential speciation events [3,4,5,6]. Presently, it is unclear whether and to what extent the identified environmental drivers of primate dispersal are scale-dependent, as in other non-primate taxa [48,49,50,51]; but see [42]. Furthermore, within-site environmental variation can have significant influence on functional genetic connectivity through impacts on the immigration and/or emigration phases of dispersal [4,52]. Despite its broad relevance to primate behavior, the extent to which within-site habitat quality impacts primate dispersal remains untested.

Here, we use a landscape genetics approach to assess potential environmental drivers of black-and-white ruffed lemur dispersal throughout southeastern Madagascar’s Ranomafana region. Ruffed lemurs (Genus *Varecia*) are an excellent taxon in which to evaluate the relationship between environmental variability and primate dispersal. Despite their reputation as obligate frugivores, ruffed lemurs are relatively ecologically flexible and inhabit forests of varying quality and structure across their latitudinal gradient [53,54,55,56,57]. However, remaining black-and-white ruffed lemurs are structured spatially and genetically into northern and southern populations [58]. Southern ruffed lemur sites (those south of the Mangoro River) are more environmentally fragmented, and the ruffed lemurs therein are more genetically isolated and less genetically diverse than those in northern populations [58]. Across their range, dispersal is facilitated primarily by available habitat cover, deterred by proximity to human settlements, and, contrary to expectations, environmental features such as rivers and altitude are unrelated to range-wide gene flow in the species [46]. It is, however, unclear whether these patterns might vary at different spatial scales (as in [48,49,50,51]). Further work has identified local habitat quality as a major predictor of ruffed lemur occupancy across the species’ range [55]; however, studies have not yet assessed the impact of source habitat quality on ruffed lemur dispersal decisions.

Despite evidence of genetic isolation in ruffed lemurs south of the Mangoro River [46], we recently found evidence of functional connectivity throughout Ranomafana National Park and the adjacent Ambositra-Vondrozo Forest Corridor (COFAV), which encompass the greatest contiguous stretch of forested habitat remaining within ruffed lemurs’ current southern range [59]. As with all remaining ruffed lemur habitats, these areas are subject to ongoing habitat transformation due to slash-and-burn agriculture (tavy), mining, and selective logging [60,61,62], leading to significant environmental heterogeneity across the region. Furthermore, within ruffed lemurs’ habitats, there is evidence of significant environmental variability in terms of forest structure and floristic diversity resulting from historic and contemporary anthropogenic activities [59,62]. This environmental variation across the landscape, combined with documented patterns of ruffed lemur connectivity throughout the southeastern rainforest corridor, makes the Ranomafana National Park and the adjacent COFAV region an excellent location to evaluate the role of environmental variation on patterns of short-range dispersal in this species. 

Given ruffed lemurs’ reliance on large-canopied trees and high levels of frugivory [63], we expected that higher-quality sites (e.g., those with greater productivity and/or greater structural complexity) would be more attractive to dispersers than sites of lower quality (as identified in [59]). Between-site forest cover is a major driver of dispersal and gene flow in black-and-white ruffed lemurs across the species’ range [46]. We, therefore, expected that reductions in forest cover (via loss or degradation) would impede gene flow throughout the Ranomafana region. We tested for the influence of both historic and contemporary forest cover, as time lags are often present when detecting the impacts of environmental variables on genetic signatures [64]. Furthermore, as elevation closely relates to forest structure and floristic diversity worldwide [65,66,67,68,69,70,71], we also expected altitude to indirectly drive gene flow via its effect on plant communities throughout the Ranomafana region. Because environmental impacts on species are often scale-dependent [48,49,50,51], we compared our results to those from Baden et al. [46] to evaluate similarities and differences in environmental drivers of short-range dispersal (this study) and long-range dispersal/multigenerational gene flow in ruffed lemurs, as previously identified [46]. Furthermore, to our knowledge, our study is the first to evaluate the impact of within-site habitat characteristics on gene flow in primates and will inform our understanding of how local environmental characteristics influence the dispersal process, particularly the emigration and immigration phases [4]. Finally, by comparing regional and species-wide drivers of gene flow, this study strengthens our understanding of the role environmental variation plays in community structuring, and by extension, the evolutionary process at varying scales.

## 2. Materials and Methods

### 2.1. Ethics Statement

This research adhered to the American Society of Primatologists Principles for the Ethical Treatment of Non-Human Primates. The research complied with the laws and guidelines set forth by ANGAP/Madagascar National Parks and Hunter College IACUC (#AB-impact 4/18-01).

### 2.2. Genetic Sampling, Relatedness, and Differentiation

The multilocus genotype data included 10 microsatellite loci for 159 adult black-and-white ruffed lemurs (*V. variegata*; 67 males, 54 females, 38 unknown; [59]). Fecal samples were collected between 2015–2017 from 15 localities in the Ranomafana National Park and the Ambositra-Vondrozo Forest Corridor region with distances between sample localities ranging from 0.85 to 38.56 km (Figure 1). The provenance of each sample was reported to the level of an individual, where individuals sampled together were assigned the same geographic coordinates. 

Using the full dataset, we estimated pairwise relatedness using ML-Relate [72,73] and removed all pairs of individuals with a relatedness coefficient > 0.5. Our reduced sample included 62 individuals at 15 localities (mean = 4.13 individuals/locality, range = 1–10 individuals); 26 within-locality dyads (mean = 1.86 dyads, range = 1–3 dyads/locality), and 94 between-locality dyads (mean = 2.62 individuals, range = 1–8 individuals with relatives at other localities) shared a relatedness of 0.5. Removal of related individuals is generally recommended to avoid introducing artificial structure in population and landscape genetic assessments [74]. However, there is ongoing discussion regarding this approach [75,76]. While a threshold of >0.5 retained some parent-offspring and full sibling dyads (i.e., those with r = 0.5) in this study, the average relatedness within (r = 0.15) and among sites (r = 0.08) was low, and our decision was ultimately a compromise between reducing potential biases introduced by including related individuals and retaining sufficient samples sizes for downstream analyses. 

Using the remaining individuals (N = 62), we estimated individual pairwise genetic dissimilarity by calculating Rousset’s Ar [77] in the program FSTAT v.2.9.4 [78].

**Figure 1 genes-14-00746-f001:**
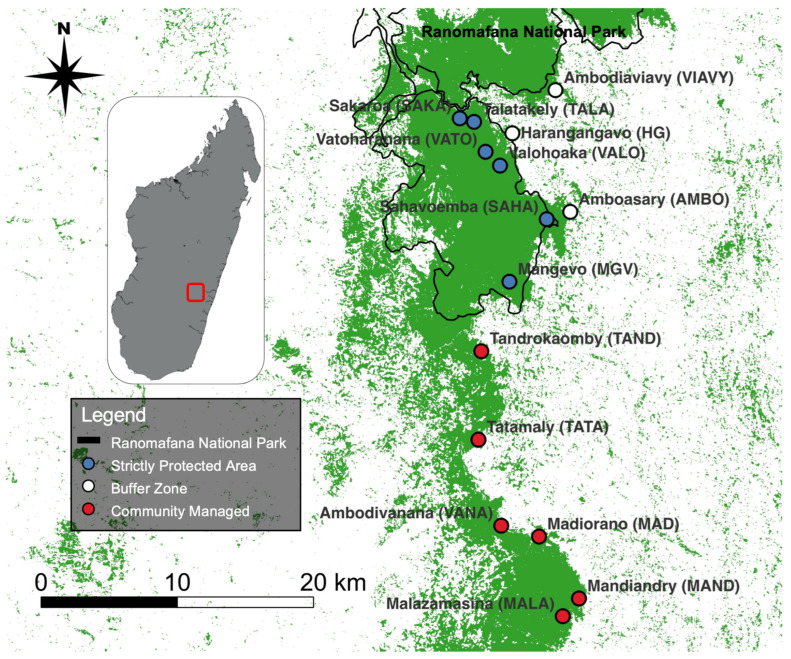
Map illustrating sampling localities (*n* = 15) across Ranomafana region with protection status noted. GIS layers courtesy of Gerber [79] and Hijmans et al. [80].

### 2.3. Landcover Classification and Landscape Feature Selection

We selected eight landscape features that were hypothesized to influence ruffed lemur movement and gene flow: (1) 1990 forest cover (historic); (2) 2016 forest cover (contemporary); (3) rivers; (4) altitude; (5) topographic position index (TPI); (6) terrain ruggedness index (TRI); (7) fire density; and (8) canopy height (Figure 2). Categorical surfaces analyzed included 1990 and 2016 forest cover and rivers, while continuous surfaces included altitude, TPI, TRI, fire density, and canopy height. Forest cover in 1990 and 2016 were classified using spectral mixture analysis (SMA) and linear spectral unmixing (see Appendix A for details). River data were downloaded from Open Street Map (https://www.openstreetmap.org/; accessed on 5 January 2021) as vector data and were rasterized. Altitude data from the Shuttle Radar Topography Mission (SRTM) were downloaded from the USGS Earth Explorer data portal (https://earthexplorer.usgs.gov/; accessed on 5 January 2021) at 1 arc second (approximately 30 m) resolution. Topographic position index, a measure that compares the elevation of each pixel in the raster to the adjacent landscape and calculates a quantitative value that is indicative of the pixel’s relative position (i.e., slope, valley, plain, or ridge), was derived from altitude data using the topographic position index function from the GDAL library (http://www.gdal.org/; accessed on 5 January 2021) in QGIS v.3.20. Terrain ruggedness index, a measure of the ruggedness of a pixel calculated by comparing elevation differences between a pixel and its eight neighboring cells within the raster [81], was also derived from the altitude data using the terrain ruggedness index function from the GDAL library (http://www.gdal.org/; accessed on 5 January 2021) in QGIS v.3.20. Fire data were downloaded from the NASA Fire Information for Resource Management System (FIRMS) from both the MODIS Collection 6 (MC6; 2001–2016) and the VIIRS S-NPP 375 m (2012–2016) datasets. Data from MC6 and VIIRS were subsampled to only include nominal and high-confidence fire reports (above 30% confidence; [82]), data were merged, and fire density was estimated using the Heatmap Kernel Density Estimation function in QGIS v.3.20 with a radius of 500 m (based on suggestions from Page et al. [83]). ‘No-data’ pixels were filled with a value of 0 to indicate no fires in that pixel. We downloaded 2019 (contemporary) canopy height data from Potapov et al. [84], which were generated through the integration of the NASA Global Ecosystem Dynamics Investigation (GEDI) spaceborne lidar system on the International Space Station and a timeseries of Landsat imagery. We resampled canopy height data to a 10,000 m^2^ resolution and corrected non-forest values (over 60) to 0. All surfaces were converted to a uniform geographic coordinate system (Universal Transverse Mercator; UTM), resampled at a resolution of 10,000 m^2^, and clipped to the study extent for analysis. Previous work has shown that changes in spatial resolution do not significantly alter the results of landscape genetic analyses [85,86]. Therefore, this resolution was chosen as a trade-off between retaining detail across the landscape and minimizing processing time for analyses. Straight-line Euclidean distance between individuals (i.e., geographic distance) was not included as an additional factor in our models, as distance is incorporated as a null model in ResistanceGA. We did, however, conduct a linear regression between pairwise Euclidean and genetic (Ar) distances to identify the proportion of our data influenced by geographic distance alone.

### 2.4. Resistance Surface Parameterization and Optimization

Resistances between sampling localities were calculated using the commuteDistance function from the R package gdistance [87], based on average pairwise resistances using an eight-neighbor connectivity scheme, and optimized using the R package ResistanceGA [88]. ResistanceGA utilizes genetic algorithms to adaptively search a broad parameter space to determine the optimal resistance values that best describe pairwise genetic differentiation (in our study, Ar). This approach makes no a priori assumptions about the direction or magnitude of the resistance between landscape and genetic distances, allowing for a more thorough investigation of the relationship between landscape features and gene flow than more widely used methods (i.e., expert-based value assignments; [89,90]). Continuous surfaces were optimized using Monomolecular and Ricker transformations, while categorical surfaces were optimized by holding one feature constant at a value of 1 and then adjusting resistance values for all other features between the values of 0 and 5000 for single surfaces and 10,000 for composite surfaces. The analyses were run in parallel on the Extreme Science and Engineering Discovery Environment (XSEDE) Bridges-2 Regular Memory platform [91] using a modified Singularity environment from Finnish Center for Science (CSCfi; https://github.com/CSCfi/singularity-recipes; accessed on 11 March 2021). We evaluated the resistance optimization process for each surface (i.e., landscape feature) using log-likelihood (corroborated with AICc; Akaike’s Information Criterion corrected for small/finite population size; [92]), which was determined from linear mixed-effects models with MLPE parameterization [93] and evaluated by maximum likelihood in lme4 [94].

### 2.5. Resistance Surface Model Selection

To account for our uneven sampling design and control for bias, we conducted bootstrap resampling using 75% of the data (47 sampling locations) following Ruiz-Lopez et al. [47]. These data were randomly selected without replacement, and then optimized surfaces were fit to the selected data. Following 10,000 iterations, the average rank and average model weight ($\bar{\omega }$) were determined for each resistance surface, along with the frequency with which a surface was ranked as the top model ($\hat{\pi }$) in order to address uncertainty in the top model; Burnham & Anderson [95] identify $\hat{\pi }$ as the bootstrap equivalent of ω. After identifying the top surfaces in isolation, we ran Spearman’s rank correlation between the surfaces to assess the degree of correlation. We created and optimized composite surfaces by combining the top models; surfaces that had both a greater selection frequency ($\hat{\pi }$) than distance alone and were selected more than one percent of the time ($\hat{\pi }$ > 1.00) were used to generate composite surfaces. All single and composite surface optimization processes were conducted at least twice, as per recommendations by Peterman [96], to ensure convergence on the top model(s). Following optimization, we again conducted a bootstrap model selection using 10,000 iterations, and average rank, ω, and $\hat{\pi }$ were calculated to assess composite models in relation to their component surfaces. Finally, current flow across the landscape was visualized in Circuitscape v4.0.5 using the best supported resistance surface(s). Landscape surfaces generated during the current study are available on GitHub (https://github.com/amandamancini/ruffed_lemur_landscape_genetics accessed on 5 January 2021), along with the Singularity image and all code used to parameterize, optimize, and assess resistance layers.

### 2.6. Gravity Model

To evaluate how within-site conditions influence dispersal and functional population connectivity, we used singularly constrained gravity models based on a saturated network. Gravity models use a network-based approach composed of nodes and edges to evaluate both within- and between-site environmental drivers of functional population connectivity, respectively [97]. To evaluate within- and between-site drivers, gravity models integrate three parameters: the distance between sites (ω), influence of within-site conditions on attraction of individuals to or from a site (υ), and the resistance of intervening landscape features between sites (*c*). We used proportion of shared alleles (D_PS_) as our measure of gene flow as this metric is free of equilibrium assumptions and can represent shared information, therefore satisfying the “maximum entropy information minimizing” approach used in landscape genetic gravity models (for more details see Murphy et al., [98]). We calculated D_PS_ using the R package adegenet and implemented the singularly constrained gravity models in R in package GeNetIt [37]. 

We used Euclidean distance between dyads (ω) and landscape resistance (c) calculated from optimized resistance assignments in ResistanceGA [96]. Within-site (υ) variables included forest structure and floristic diversity measures, average Normalized Difference Vegetation Index (NDVI), and topographic variation. Forest structure measures included average canopy height, basal area, and stem density at each sampling site, and floristic diversity included effective number of species (ENS) and average Importance Value Index (IVI) of ruffed lemur food trees at each site. Both forest structure and floristic diversity were quantified from three to five 25 m by 25 m plots at each site (described in Mancini [59]). Normalized Difference Vegetation Index (NDVI) was used to evaluate site vegetation density and was calculated as (NIR − VIS)/(NIR + VIS) from the 2016 Landsat OLI imagery, and mean NDVI was calculated within each of the 15 sampling sites. Topographic variation was quantified as the standard deviation of altitude using data from the Shuttle Radar Topography Mission (SRTM) within each of sampling sites. Site extents for mean NDVI and topographic variation were defined by a 500-m buffer around each botanical plot noted above. Within-site conditions (υ) for each individual were assigned based on the site in which an individual was sampled, where all individuals sampled within a site were assigned the same within-site conditions. 

We initially evaluated nine ‘within-site’ models against a null model of isolation-by-distance (IBD) to evaluate the significance of within-site conditions alone in explaining current ruffed lemur gene flow. Within-site models included singular models for each of the seven within-site conditions, forest structure (a combination of canopy height, basal area, and stem density), and forest productivity (a combination of ENS, IVI, and NDVI). We combined within-site variables that performed better than IBD with resistance distance variables to evaluate the joint potential of within- and between-site metrics in explaining ruffed lemur gene flow. In total, we tested 15 composite models along with their component singular models. We log-transformed the response variable (D_PS_; gene flow) and all predictor variables. Gravity models were performed using linear mixed effect models fit using Restricted Maximum Likelihood (REML), and model selection was conducted using log-likelihood (corroborated with Akaike Information Criterion; AIC). Finally, we used β estimates from all singular models as an evaluation of the variables’ directional effect on connectivity, with positive β values suggesting within-site characteristics were attractive to ruffed lemur dispersal [99].

## 3. Results

### 3.1. Resistance Analysis 

We found a significant signature of isolation-by-Euclidean distance, which explained 4.4% (Pearson’s r = 0.2091) of the observed population genetic structure (Appendix A). Despite evidence of IBD, three surfaces—terrain ruggedness index (TRI), canopy height, and topographic position index (TPI)—were more strongly associated with genetic differentiation than geographic distance alone (Table 1, Table 2, Table 3 and Appendix A). Resistance to terrain was lowest in areas of low-to-moderate ruggedness where altitude did not vary greatly on a fine scale (below 10 ha; Figure 3 and Table 2). Relatively tall canopy (above 15 m) appears to better facilitate functional connectivity and gene flow than areas with low canopy height (Figure 4 and Table 2). Furthermore, resistance was lowest on flat or moderate slopes leading to ridges (Figure 5 and Table 2). The remaining five surfaces (1990 and 2016 forest cover, altitude, rivers, and fire density) explained slightly more variation than distance alone but were seldom chosen as the top model (less than 1.0% of the time) and thus were not considered in the composite analysis (Table 1). Although both 1990 and 2016 forest cover were rarely considered in the top model, results for both single surface resistance models indicated that ruffed lemur gene flow was more resistant through matrix than forest (Table 3). Additionally, resistance decreased monotonically with increasing altitude and exponentially with increasing fire density (Appendix A and Table 2). Finally, rivers were predicted to cause more resistance to gene flow in ruffed lemurs than the intervening landscape (Table 3).

To assess the combined effects of terrain, canopy height, and topography, we created four composite surfaces that combined TRI, canopy height, and TPI together in all permutations and compared results against the isolated surfaces, as well as against straight-line Euclidean distance. There was low correlation among all surfaces used in the composite models, with Spearman rank correlation coefficients less than 0.10 in all cases (Appendix A). There was no clear consensus on the top model predicting ruffed lemur gene flow, as both TRI and canopy height in singular, the composite of TRI and canopy height (Combination C), and the composite of TRI and TPI (Combination D) were all ranked similarly and were top models more than 10% of the time (Table 4, Table 5 and Appendix A). Resistance patterns in the composite surfaces showed an identical pattern for TRI as with the variable in singular, where resistance was lowest in areas of low-to-moderate ruggedness (Figure 6 and Figure 7). Canopy height was transformed to distance when combined with TRI, meaning this composite surface was essentially identical to TRI in isolation (Combination C; Figure 6). However, for the other two composite models containing canopy height (Combinations A and B), we found the same pattern as canopy height in singular, where taller trees above 15 m appear to facilitate dispersal (Appendix A). Transformations for TPI showed greater variability, although for composite surfaces containing TPI resistance was lowest on flat or moderate slopes leading to ridges, as was found in the singular surface (Figure 7, Appendix A). 

For both highly ranked composite surfaces (Combinations C and D), TRI contributed to more than 95% of the surface, and results were therefore driven primarily by TRI (Appendix A). This suggests that TRI on its own was the top model approximately 60% of the time (a sum of TRI in singular and contributions from Combinations C and D), lending support for TRI as the best predictor of the genetic data. However, canopy height was also a top-ranked model nearly 20% of the time and, therefore, cannot be excluded as a strong predictor (Table 4). Our visualization of the optimized TRI model shows a high degree of current (gene) flow throughout the eastern extent of our study area, particularly in the eastern portion of the remaining forest (Figure 8). Our visualization of the canopy height model revealed a relatively uniform current flow throughout much of the remaining rainforest in our study extent, although the visualization does suggest current flow may be greatest in the eastern and western boundaries of the remaining forest (Figure 9). Together, these results suggest a high degree of connectivity throughout all localities sampled, although the northern-most site (VIAVY) seems to have weaker current flow than any of the other sites (Figure 8 and Figure 9). 

### 3.2. Gravity Model

When investigating within-site drivers of dispersal, we found equal support for the null model of IBD and the test models of IVI and NDVI, with no support for the other test models (Table 6). Unsurprisingly, we found a negative relationship between distance and D_PS_, suggesting fewer shared alleles between individuals at increasing geographic distances (β = −0.030, SE = −0.003, df = df = 3685, t = −9.36, *p* < 0.001). We found a significant and slightly positive relationship between the average IVI of *Varecia* food resources and D_PS_, suggesting that these food resources were attractive to ruffed lemur dispersers (β = 0.093, SE = 0.036, df = 60, t = 2.62, *p* = 0.011). Finally, we found a negative (but non-significant) relationship between mean site NDVI and D_PS_, suggesting that sites with higher productivity values have a reduced attractiveness to dispersal than sites with lower NDVI values (β = −0.261, SE = 0.465, df = 60, t = −0.56, *p* = 0.576). Together, these results indicate that sites such as TALA, SAKA, and HG are more likely to attract dispersers due to their relatively higher values of average IVI and lower values of average NDVI compared to sites such as MGV, MAND, and MALA, which display lower average IVI and higher average NDVI (Figure 1 and Appendix A).

Finally, we combined the best-supported within-site models (IVI and NDVI) with the two best-supported resistance models (TRI and DSM) to evaluate the combined influence of within- and between-site drivers on ruffed lemur dispersal. First, all models containing one or both resistance variables were more strongly supported than the null model of IBD (Table 7). Two models—canopy height in singular and canopy height combined with NDVI—showed equal support as the best models predicting dispersal in ruffed lemurs throughout the Ranomafana region (Table 7). As expected, in the composite model, we found a significant and negative relationship between canopy height resistance distance and D_PS_, confirming that greater resistance distances reduced gene flow (β = −0.216, SE = 0.024, df = 3683, t = −9.01, *p* < 0.001). Furthermore, we found a negative (but non-significant) relationship between mean site NDVI and D_PS_, similar to the variable in singular (β = −0.488, SE = 0.437, df = 60, t = −1.12, *p* = 0.269). As with the composite model, we found a significant and negative relationship between canopy height resistance distance and D_PS_ when evaluating the variable in singular (β = −0.215, SE = 0.024, df = 3683, t = −8.96, *p* < 0.001). 

## 4. Discussion

The main objective of this study was to evaluate the between- and within-site environmental drivers of black-and-white ruffed lemur dispersal throughout Madagascar’s southeastern Ranomafana region. We found that the predominant between-site drivers of ruffed lemur dispersal were terrain ruggedness index (TRI) and canopy height, where areas of low-to-moderate ruggedness and relatively tall canopy heights (above 15 m) facilitated functional connectivity and gene flow. The other six environment variables evaluated (topographic position index, 1990 and 2016 forest cover, altitude, rivers, and fire density) were not significant predictors of ruffed lemur dispersal, although resistance was lowest within forests, at moderate slopes, at lower altitudes, in areas with lower fire density, and outside of rivers. Productivity metrics, including the Normalized Difference Vegetation Index (NDVI) and the average Importance Value Index (IVI) of ruffed lemur food trees, were the most influential within-site drivers of ruffed lemur dispersal, with animals being more likely to disperse into areas with lower NDVI and higher IVI values. However, when between- and within-site drivers were combined, models containing only between-site resistances were consistently the best supported. We, therefore, conclude that the within-site environmental variables tested were less influential in ruffed lemur dispersal decisions than between-site factors. 

### 4.1. Between-Site Influences on Ruffed Lemur Dispersal

The best-supported between-site drivers of ruffed lemur dispersal in the Ranomafana region were terrain ruggedness index (TRI) and canopy height. Ruffed lemur dispersal was facilitated by areas of low-to-moderate ruggedness and impeded by increasingly rugged terrain. Our results add to a growing body of evidence identifying areas of high ruggedness as impediments to gene flow (including lizards, *Sceloporus occidentalis*: [100]; stone marten, *Martes foina*: [101]; Northern quoll, *Dasyurus hallucatus*: [102]; and tigers, *Panthera tigris*: [31]; but see Balkenhol et al. [49], wolverine, *Gulo gulo*). For instance, tigers preferentially selected the least rugged dispersal paths from within their preferred landscapes—forested areas with low anthropogenic presence—suggesting that avoiding areas of high terrain ruggedness was secondary to moving through their preferred landscape [31]. We found a similar pattern in ruffed lemurs, with gene flow predicted to be highest in areas of low-to-moderate ruggedness in the eastern boundary of the remaining forest of the Ranomafana region (Figure 3 and Figure 8). Interestingly, our results predicted that ruffed lemurs would avoid areas of the lowest ruggedness (Figure 3). The reasoning for this may be twofold: low ruggedness areas of the eastern Ranomafana region are associated with increased anthropogenic presence and are preferentially used by humans for both irrigated and shifting agriculture (i.e., flooded rice paddies and slash-and-burn or tavy; [103,104]), whereas low ruggedness areas in the western region are associated with higher altitudes and shorter forest canopies [65,66,67]. In this way, similar to tigers, the ruffed lemurs in our study area are likely dispersing through the least rugged terrain that enables them to avoid anthropogenic landscapes while also remaining within their preferred tall-canopied habitats [31]. Alternatively, the relationship found between topography and dispersal may be driven by ruffed lemur behavioral ecology, similar to what has been hypothesized in wolverines [49]: ruffed lemurs segment their territories along ridgelines (Baden, pers. comm.), making it possible that individuals preferentially move along ridgetops with less rugged terrain to avoid core territories of unknown individuals. 

In addition to terrain ruggedness, canopy height was also predicted as a significant driver of ruffed lemur dispersal in the Ranomafana region. We found that relatively tall canopy (above 15 m) facilitated and short canopy height impeded dispersal between localities. In other forest-dwelling species, canopy height or structural homogeneity were also found to be significant predictors of gene flow (Western capercaillie, *Tetrao urogallus*: [105]; black-eared mouse, *Peromyscus melanotis*: [106]), although unsurprisingly the forest-dwelling scarlet macaws (*Ara macao*)—which disperse via flight—were not impacted by either [107]. Ruffed lemurs rely on tall, broad-canopied fruiting trees due to their obligate frugivory [53,56] and, despite their ecological flexibility, are found at greater densities in localities characterized by these features [57,108,109]. It is thus unsurprising that taller forest canopy facilitates dispersal, as animals tend to preferentially move through landscapes that are most similar to their preferred habitat [11,12]. The high rates of gene flow identified by our study likely correspond with areas of tall canopy at lower altitudes along the eastern boundary of the remaining forest in the Ranomafana region, similar to patterns found in other parts of the world [65,66,67] (Figure 9). Our results also predicted high rates of gene flow throughout fragmented forests along the western edge of the region (Figure 9); however, dispersal through these areas is unlikely given its higher altitude, as there are few records of ruffed lemurs occurring above 1200 m and no evidence above approximately 1350 m [110,111,112]. 

Curiously, forest cover was not identified as a significant driver of ruffed lemur dispersal, as in Baden et al. [46], which may reflect scale-dependent differences in dispersal drivers. Specifically, the simple presence of forest may be an important driver of gene flow for long-range and multigenerational dispersal, while the quality of forest may be most influential for short-range dispersal decisions. Alternatively, these results may simply mirror those found by Milanesi et al. [105], where forest structure derived from LiDAR remote-sensing yielded better estimates of gene flow compared to traditional land cover data.

### 4.2. Within-Site Influences on Ruffed Lemur Dispersal

Ultimately, within-site habitat features in this study were less influential than between-site features in driving ruffed lemur dispersal in the Ranomafana region. These findings are similar to other recent studies using similar methodologies (black-capped vireos, *Vireo atricapilla*: [112]; Arizona treefrog, *Dryophytes* (*Hyla*) *wrightorum*: [113]). Although not significant, productivity measures– Normalized Difference Vegetation Index (NDVI) and the average Importance Value Index (IVI) of ruffed lemur foods– were the best supported habitat features driving ruffed lemur emigrations and immigrations. Sites with relatively lower IVI and higher NDVI, such as Mangevo (MGV), Mandiandry (MAND), and Malazamasina (MALA), likely act as source populations (i.e., source of emigrants), and those with relatively high IVI and lower NDVI, such as Talatakely (TALA), Sakaroa (SAKA), and Harangangavo (HG), are likely most attractive to dispersers and may act as sinks (Figure 1; Appendix A). This finding is further supported by the presence of several first-generation immigrants identified in both TALA and SAKA, suggesting that these sites are indeed attractive to dispersers [59]. 

There is high within-site structural variability between the sampling localities evaluated in this study, with more pristine and higher quality sites such as Mangevo (MGV), Tandrokaomby (TAND), and Amboasary (AMBO) having taller canopies, larger basal area, and greater stem density than disturbed, lower quality sites of Talatakely (TALA) and Sakaroa (SAKA; Appendix A) [59]. Many forest-dwelling species prefer to move through higher quality, complex forest stands, as opposed to open habitat [114,115,116], and preferentially incorporate these stands into their home range [116,117,118]. Furthermore, several studies have found that higher quality sites (relative to the species’ biology) facilitated gene flow and may act as sources of emigrants for the species (American martens, *Martes americana*: [52]; Columbia spotted frogs, *Rana luteiventris*: [98]; Blainville’s horned lizard, *Phrynosoma blainvillii*: [119]). It was, therefore, surprising that within-site forest structure was not identified as a significant predictor of ruffed lemur dispersal. It is possible that our metrics did not fully capture details of forest structure that are important to ruffed lemur movement. In the future, LiDAR or photogrammetry could enable a more detailed quantification of forest structure that our study may have missed, while allowing researchers to measure other important characteristics such as canopy density and size and distribution of canopy gaps [115,119,120,121].

### 4.3. Spatial Variation in Response to Environmental Variables

In addition to our evaluation of between- and within-site drivers of dispersal, we compare our regional results to a recent range-wide assessment by Baden et al. [46] to evaluate how ruffed lemur dispersal might vary across spatial scales. The study by Baden et al. [46] assessed pattens of long-range dispersal and multigenerational gene flow and found that long-range ruffed lemur dispersal is facilitated primarily by available forest cover. By contrast, the present study evaluated smaller-scale, more regional gene flow from typical dispersal events and found weak evidence for this same relationship. Forest cover is contiguous throughout our study area, making it possible for ruffed lemurs to avoid the matrix when dispersing through the region, perhaps explaining why 1990 and 2016 forest do not appear to play significant roles in regional ruffed lemur dispersal. We did, however, find the height of contemporary forest—a strong correlate of 2016 (contemporary) forest cover (ρ = 0.77; Table 6) and a moderate correlate of 1990 (historic) forest cover (ρ = 0.44; Appendix A)—was a significant predictor of dispersal, with taller canopy facilitating dispersal. Tall canopy in the Ranomafana area is often related to higher quality forest [60,65] where ruffed lemurs are found at their highest abundances [53,108,109]. Additionally, ruffed lemurs rely on large, broad canopy trees for quadrupedal movement, access to sufficient fruit [53], and successful reproduction [122]. Therefore, ruffed lemurs’ utilization of tall canopy trees for dispersal is expected, particularly as animals preferentially use areas most similar to their chosen habitat for dispersal [11,12]. 

Proximity to human settlements was found to be the main deterrent to long-range ruffed lemur dispersal and multigenerational gene flow [46]. In comparison, we did not find a significant influence of fire density—a more nuanced metric of anthropogenic presence—on short-range ruffed lemur dispersal. Although not significant, we did find an exponential increase in resistance with increasing fire density, similar to the pattern found by Baden et al. [46] with proximity to human settlements. Topography was not found to be a significant driver of long-range ruffed lemur dispersal across their remaining range [46], although we did find the terrain ruggedness index to be a significant between-site driver of short-range dispersal in the Ranomafana region, with low-to-moderate ruggedness facilitating dispersal. Finally, neither study found any influence of rivers on dispersal, despite the role rivers have played in the biogeography of Malagasy primates [110,123]. The permeability of rivers is likely dependent on several factors, including size, flow, and access to bridging structures (e.g., overhanging trees or anthropogenic structures such as bridges), with larger waterways expected to be less permeable than minor rivers [124,125]. Only minor waterways are present within our study and, therefore, unlikely to pose dispersal barriers to the species, as they are likely able to disperse across these minor rivers via natural canopy bridges.

### 4.4. Conclusions

In summary, we found that between-site environmental characteristics– specifically, terrain ruggedness and forest canopy height– were better predictors of short-range ruffed lemur dispersal than within-site characteristics such as NDVI and Importance Value Index of ruffed lemurs’ foods in the Ranomafana region. Furthermore, we found evidence of scale-dependent environmental influences on ruffed lemur dispersal, with topography only driving short-range dispersal decisions. Forest characteristics influenced dispersal at all scales, with forest presence facilitating multigenerational, long-range dispersal and taller forest heights facilitating short-range dispersal in ruffed lemurs. Together, results from this and earlier studies [46] highlight the importance of high-quality forest to sustaining ruffed lemur gene flow across spatial scales. Given the accelerating forest modification, loss, and fragmentation throughout Madagascar’s eastern rainforest escarpment [55,112,126,127,128,129], it is likely ruffed lemur dispersal, along with the dispersal patterns of countless other arboreal taxa [130], will become increasingly restricted in the future. Dispersal capacity and limitations are primary drivers of community structuring across primate taxa at regional spatial scales, more so than niche differentiation [3,130]. If ruffed lemur dispersal becomes significantly limited by forest loss, this may lead to more divergent primate community structure, even between neighboring localities [130]. Shifting community structure, decreasing forest suitability, and limited dispersal ability will have cascading effects, and will likely result in massive decreases in population sizes beyond what would be expected with shifting suitable habitat alone [131,132]. Therefore, retaining high-quality forests (particularly in areas without strict protection) and forest connectivity is paramount to retaining gene flow of ruffed lemurs within their southern range, ultimately buffering against ongoing population declines and local extinctions in this critically endangered species.

## Figures and Tables

**Figure 2 genes-14-00746-f002:**
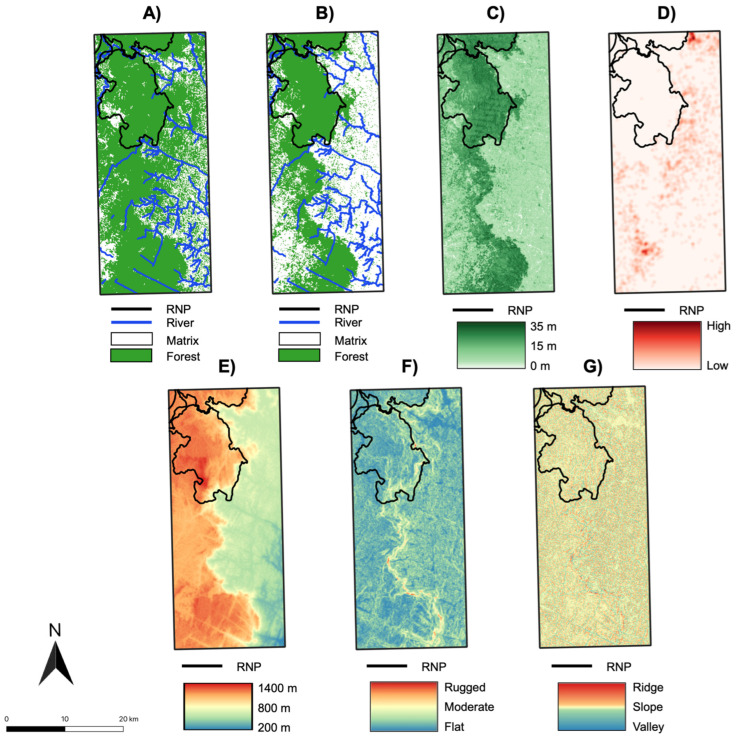
Maps of Ranomafana region showing landscape surfaces for (**A**) 1990 (historic) forest cover and rivers; (**B**) 2016 (contemporary) forest cover and rivers; (**C**) 2019 canopy height; (**D**) fire density; (**E**) altitude; (**F**) terrain ruggedness index (TRI); and (**G**) topographic position index (TPI).

**Figure 3 genes-14-00746-f003:**
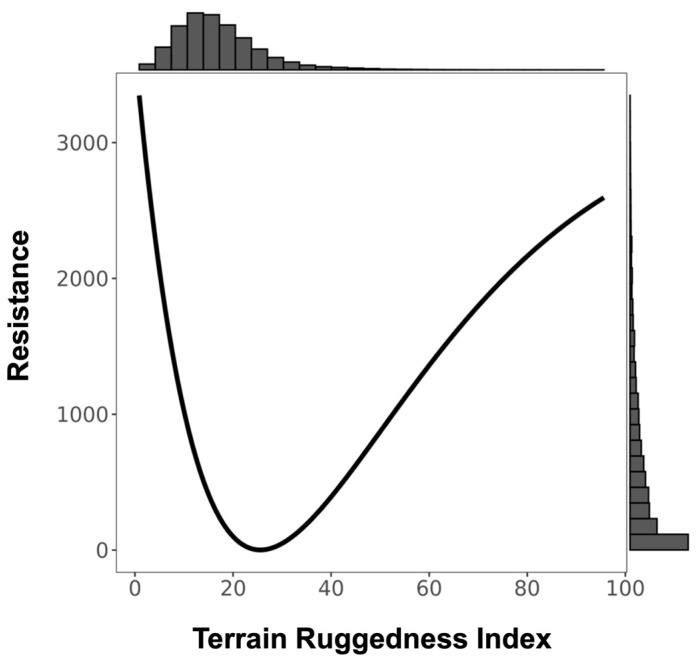
Resistance transformation plot for single surface Terrain Ruggedness Index (TRI). Low TRI values represent relatively flat terrain and high values represent rugged areas. Resistance to ruffed lemur movement was greater in flat (low value) and rugged (high value) terrain.

**Figure 4 genes-14-00746-f004:**
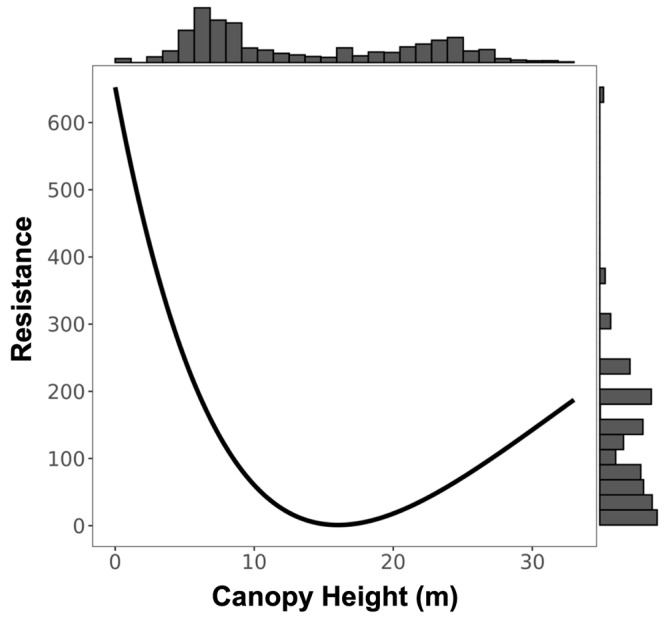
Resistance transformation plot for single surface canopy height. Resistance to ruffed lemur movement was greater in areas with canopy cover under 5 m.

**Figure 5 genes-14-00746-f005:**
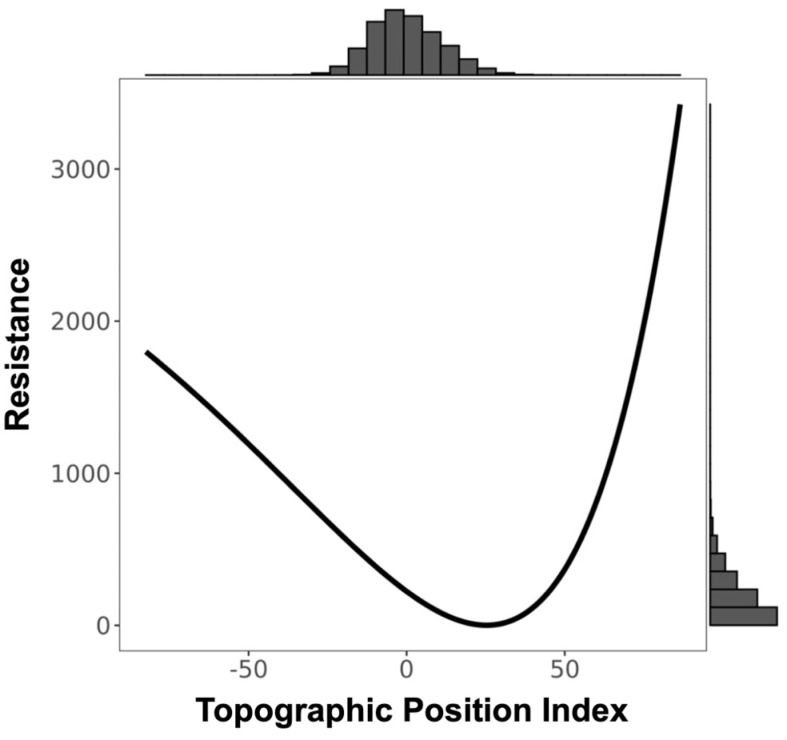
Resistance transformation plot for single surface Topographic Position Index (TPI). Positive TPI values represent ridges, negative TPI values represent valleys. Flat terrain or areas of constant slope are represented by a TPI value near zero. Resistance to ruffed lemur movement was lower for shallow slopes leading to ridges.

**Figure 6 genes-14-00746-f006:**
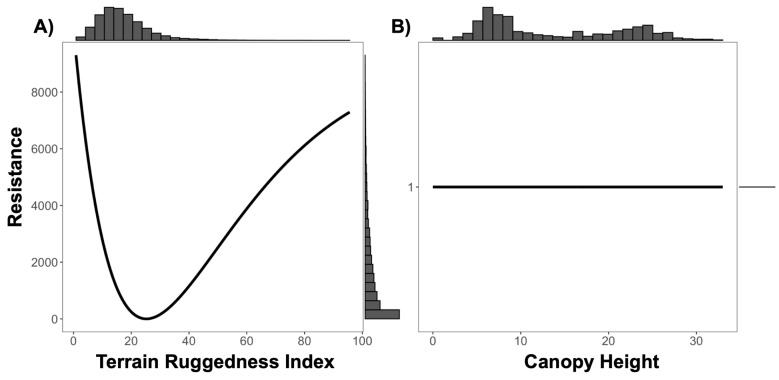
Resistance transformation plot for ‘Composite C’ surface of (**A**) Terrain Ruggedness Index (TRI) and (**B**) canopy height. Low TRI values represent relatively flat terrain and high values represent rugged areas. Resistance to ruffed lemur movement was greater in flat (low value) and rugged (high value) terrain as show in panel (**A**). Canopy height (**B**) was transformed to Distance and, therefore, was non-influential on ruffed lemur movement when evaluating ‘Composite C’ resistance surface.

**Figure 7 genes-14-00746-f007:**
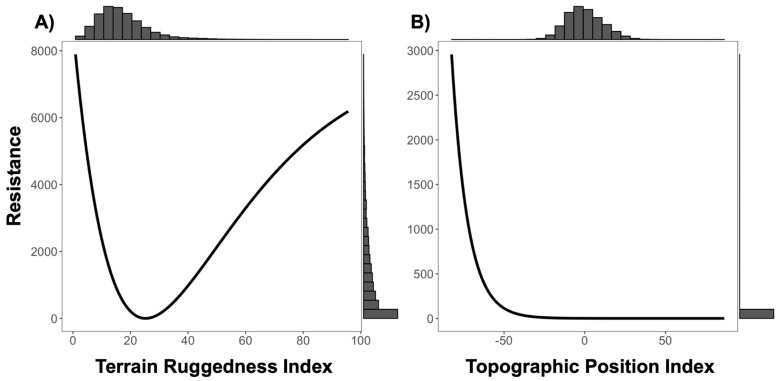
Resistance transformation plot for ‘Composite D’ surface of (**A**) Terrain Ruggedness Index (TRI) and (**B**) Topographic Position Index. Low TRI values represent relatively flat terrain, and high values represent rugged areas. Positive TPI values represent ridges, negative TPI values represent valleys, and flat terrain or areas of constant slope are represented by a TPI value near zero. Resistance to ruffed lemur movement was greater in flat (low value) and rugged (high value) terrain, as shown in panel (**A**). Resistance to ruffed lemur movement was lower for slopes leading to ridges and ridges, as shown in panel (**B**).

**Figure 8 genes-14-00746-f008:**
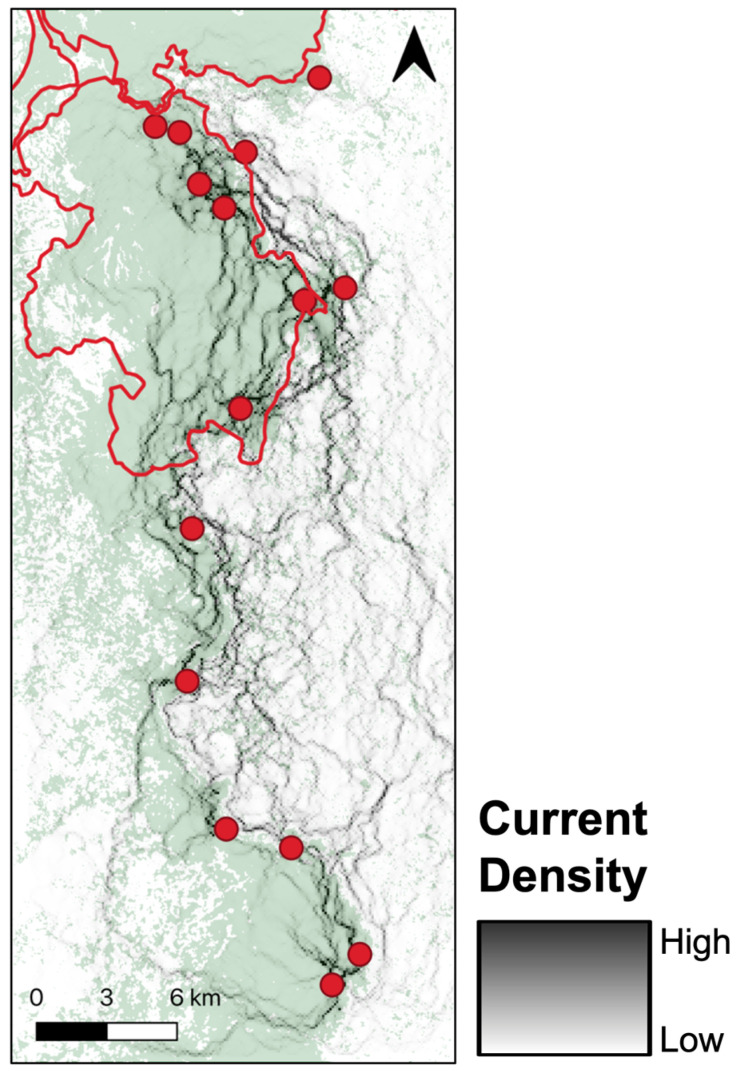
Cumulative resistance among individuals for optimized Terrain Ruggedness Index (TRI) created using Circuitscape v4.0.5 (https://circuitscape.org; accessed on 22 January 2019). Darker greys indicate areas of higher conductance (i.e., low resistance, high dispersal); lighter greys indicate areas of higher resistance (i.e., low conductance, low dispersal). Cumulative resistance is overlayed on 2016 forest cover and red circles represent the 15 sampling localities throughout the region.

**Figure 9 genes-14-00746-f009:**
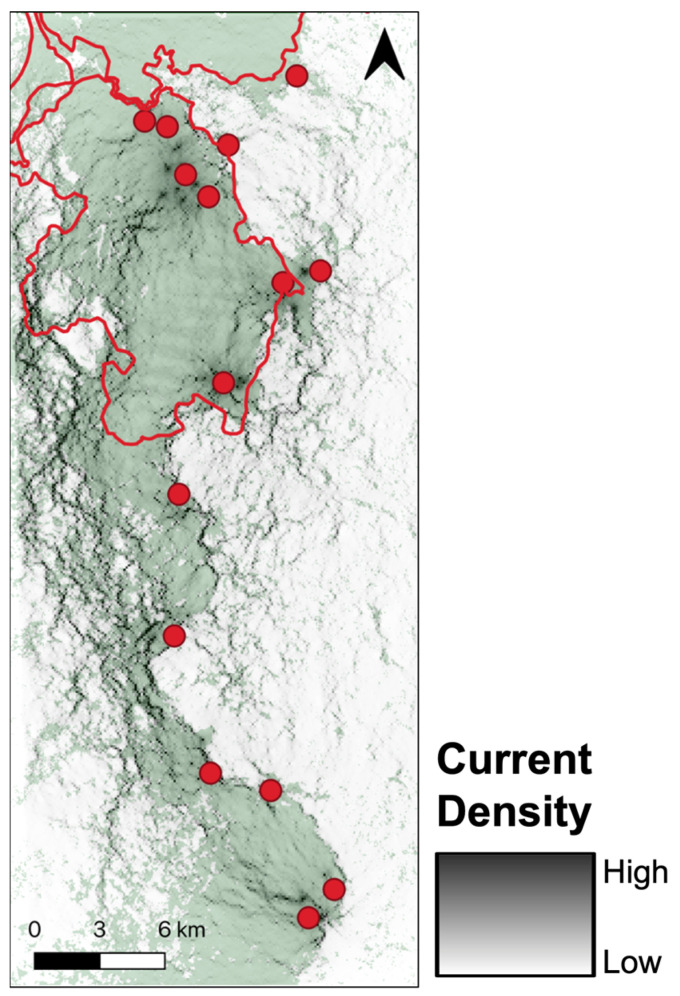
Cumulative resistance among individuals for optimized canopy height created using Circuitscape v4.0.5 (https://circuitscape.org; accessed on 22 January 2019). Darker greys indicate areas of higher conductance (i.e., low resistance, high dispersal); lighter greys indicate areas of higher resistance (i.e., low conductance, low dispersal). Cumulative resistance is overlayed on 2016 forest cover and red circles represent the 15 sampling localities throughout the region.

**Table 1 genes-14-00746-t001:** Results from bootstrap selection of optimized linear-mixed effects models on single surfaces. Rows shown in bold indicate models that performed better than Euclidean distance alone and were selected as the top model more than 1% ($\hat{\pi }$ ≥ 1.00) during 10,000 bootstrap iterations.

Layer	k	Avg. Rank	$\bar{\omega }$	$\hat{\pi }$
**TRI ^†^**	**4**	**2.278**	**0.285**	**63.34**
**Canopy Height**	**4**	**2.539**	**0.097**	**25.20**
**TPI ^‡^**	**4**	**3.671**	**0.053**	**9.42**
1990 Forest Cover	3	5.436	0.104	0.38
2016 Forest Cover	3	5.614	0.105	0.96
Altitude	4	6.064	0.027	0.30
Rivers	3	6.125	0.082	0.01
Fire Density	4	6.199	0.025	0.39
Distance	2	7.078	0.223	0.00

k = number of parameters following continuous surface transformation or number of categories in categorical surfaces; Avg. rank = average model rank following 10,000 bootstrap iterations; $\bar{\omega }$ = average model weight averaged over 10,000 bootstrap iterations, representing the probability that the model is the best of the set; $\hat{\pi }$ = proportion of bootstrap iterations in which model was chosen as the top model; † TRI: Terrain Ruggedness Index; ‡ TPI: Topographic Position Index.

**Table 2 genes-14-00746-t002:** Model results and resistance transformation variables, including the transformation equation, shape, and maximum value for the optimized continuous single surface. For each surface, the model was parameterized using log-likelihood, and AICc is presented for corroboration.

Surface	Log-Likelihood	AICc	Equation	Shape	Maximum
Altitude	1798.387	−3588.073	Reverse Monomolecular	7.381	5.192
Canopy Height	1800.029	−3591.357	Inverse Ricker	4.857	364.762
Fire Density	1798.126	−3587.551	Inverse-Reverse Ricker	4.972	2770.590
TPI ^‡^	1799.192	−3589.682	Inverse-Reverse Ricker	3.624	2568.438
TRI ^§^	1801.033	−3593.364	Inverse Ricker	2.606	3490.174
Distance	1797.957	−3591.710	-	-	-

^‡^ TPI: Topographic Position Index; ^§^ TRI: Terrain Ruggedness Index.

**Table 3 genes-14-00746-t003:** Model results and resistance values for features within the optimized categorical single surface. For each surface, the model was parameterized using log-likelihood, and AICc is presented for corroboration.

Surface	Log-Likelihood	AICc	Feature 1: Resistance	Feature 1	Feature 2: Resistance	Feature 2
1990 Forest Cover	1798.493	−3590.572	12.740	Matrix	1	Forest Cover
2016 Forest Cover	1798.511	−3588.066	3.185	Matrix	1	Forest Cover
Rivers	1798.280	−3590.607	1	Non-rivers	17.175	Rivers
Distance	1797.957	−3591.710	-	-	-	-

**Table 4 genes-14-00746-t004:** Results from bootstrap selection of optimized linear-mixed effects models on composite surfaces. Rows shown in bold indicate models that performed better than Euclidean distance alone and were selected as the top model more than 10% ($\hat{\pi }$ ≥ 10.00) during 10,000 bootstrap iterations.

Layer	k	Avg. Rank	$\bar{\omega }$	$\hat{\pi }$
**Combination C**	**7**	**3.045**	**0.007**	**12.15**
**TRI ^†^**	**4**	**3.104**	**0.375**	**30.65**
**Combination D**	**7**	**3.458**	**0.007**	**18.56**
**Canopy Height**	**4**	**4.347**	**0.161**	**17.53**
Combination A	10	4.450	<0.001	5.02
Combination B	7	4.568	0.002	8.84
TPI ^‡^	4	5.515	0.087	7.23
Distance	2	7.513	0.361	0.02

k = number of parameters following continuous surface transformation or number of categories in categorical surfaces; Avg. rank = average model rank following 10,000 bootstrap iterations; $\bar{\omega }$ = average model weight averaged over 10,000 bootstrap iterations, representing the probability that the model is the best of the set; $\hat{\pi }$ = proportion of bootstrap iterations in which model was chosen as the top model; ^†^ TRI: Terrain Ruggedness Index; ^‡^ TPI: Topographic Position Index.

**Table 5 genes-14-00746-t005:** Model results and resistance transformation variables, including the transformation equations, shapes, and maximum values for each continuous surface comprising the optimized composite surface. For each surface, the model was parameterized using log-likelihood, and AICc is presented for corroboration.

Surface	Log Likelihood	AICc	Canopy Height Trans	Canopy Height Shape	Canopy Height Max	TPI ^‡^ Trans	TPI ^‡^ Shape	TPI ^‡^ Max	TRI ^§^ Trans	TRI ^§^ Shape	TRI ^§^ Max
Comb. A	1799.883	−3575.451	Inverse Ricker	4.27	9250.48	Inverse-Reverse Ricker	3.08	14.23	Inverse Ricker	2.43	4422.37
Comb. B	1800.073	−3584.243	Inverse Ricker	4.82	3974.92	Ricker	0.76	5483.57	-	-	-
Comb. C	1801.071	−3582.323	Distance	2.77	5939.49	-	-	-	Inverse Ricker	2.57	9837.73
Comb. D	1801.025	−3586.255	-	-	-	Inverse Monomolecular	0.59	2957.12	Inverse Ricker	2.57	8347.75
Distance	1797.957	−3584.071	-	-	-	-	-	-	-	-	-

^‡^ TPI: Topographic Position Index; ^§^ TRI: Terrain Ruggedness Index.

**Table 6 genes-14-00746-t006:** Results for within-site gravity model predictions. Rows shown in bold indicate the best supported model based on log-likelihood criteria and corroborated using AICc.

Model	K	Log-Likelihood	AIC
**IVI ^†^**	**1**	**722.129**	**−1434.258**
**NDVI ^‡^**	**1**	**721.557**	**−1433.114**
**IBD ^§^**	**1**	**721.248**	**−1434.496**
Productivity	3	721.063	−1428.126
Canopy Height	1	719.966	−1429.932
ENS ^††^	1	719.835	−1429.671
Stem Density	1	719.619	−1429.238
Topography	1	719.373	−1428.747
Basal Area	1	719.137	−1428.275
Structure	3	717.375	−1420.750

^†^ IVI: Importance Value Index; ^‡^ NDVI: Normalized Difference Vegetation Index; ^§^ IBD: Isolation-by-distance; ^††^ ENS: Effective Number of Species.

**Table 7 genes-14-00746-t007:** Results for composite, within-site, and between-site gravity model predictions. Rows shown in bold indicate the best supported model based on log-likelihood criteria and corroborated using AICc.

Model	K	log-likelihood	AIC
**NDVI + Canopy Height Resistance**	**2**	**758.784**	**−1505.567**
**Canopy Height Resistance**	**1**	**758.069**	**−1506.137**
NDVI + IVI + Canopy Height Resistance	3	757.557	−1501.114
IVI + Canopy Height Resistance	2	757.280	−1502.560
NDVI + Canopy Height Resistance + TRI Resistance	3	756.445	−1498.889
Canopy Height Resistance + TRI Resistance	2	755.811	−1499.621
NDVI + IVI + Canopy Height Resistance + TRI Resistance	4	755.188	−1494.377
IVI + Canopy Height Resistance + TRI Resistance	3	754.942	−1495.884
NDVI + IVI + TRI Resistance	3	730.210	−1446.420
NDVI + TRI Resistance	2	730.004	−1448.008
IVI + TRI Resistance	2	729.988	−1447.977
TRI Resistance	1	729.764	−1449.529
NDVI + IVI	2	722.366	−1432.731
IVI ^†^	1	722.129	−1434.258
NDVI ^‡^	1	721.557	−1433.114
IBD ^§^	1	721.248	−1434.496

^†^ IVI: Importance Value Index; ^‡^ NDVI: Normalized Difference Vegetation Index; ^§^ IBD: Isolation-by-distance.

## Data Availability

Landscape surfaces, Singularity image, and all code are available on GitHub as https://github.com/amandamancini/ruffed_lemur_landscape_genetics accessed on 5 January 2021.

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
