# Peer review of "Terrain Ruggedness and Canopy Height Predict Short-Range Dispersal in the Critically Endangered Black-and-White Ruffed Lemur"

_genes, 2023, doi:10.3390/genes14030746_

Round 1

Reviewer 1 Report

This study presents highly detailed analyses of some environmental factors that may shape dispersal in Varecia variegata. I am not a primatologist, so I cannot evaluate the many specific aspects analyzed in this study. I highlight a few minor points that should be corrected and would consider reviewing this paper by some primatologists.

Abstract

Line 17 … variables in driving short-range black-and-white ruffed lemur (Varecia variegata) dispersal … . Indicate the study areas in the southeastern Madagascar's Ranomafana region.

 1. Introduction

Lines 61 – 95. I’m not a primatologist, but I suspect that research and studies on primate population genetics and dispersal are not so limited as stated (see: L. 61 … Research investigating large-scale primate movement has historically been limited …; L. 71 … Studies of primate landscape genetics remain limited …). I suggest reconsidering this paragraph and, perhaps, revise accurately the relevant literature

2. Materials and Methods

What kind of samples ...  were collected between 2015-2017 from 15 localities … ?

Author Response

Response to Reviewer 1 Comments

Point 1: Line 17 … variables in driving short-range black-and-white ruffed lemur (Varecia variegata) dispersal … . Indicate the study areas in the southeastern Madagascar's Ranomafana region.

 Response 1: “in southeastern Madagascar’s Ranomafana region” was added at line 26.

Point 2: Lines 61 – 95. I’m not a primatologist, but I suspect that research and studies on primate population genetics and dispersal are not so limited as stated (see: L. 61 … Research investigating large-scale primate movement has historically been limited …; L. 71 … Studies of primate landscape genetics remain limited …). I suggest reconsidering this paragraph and, perhaps, revise accurately the relevant literature

Response 2: To clarify our meaning in the first note of the comment, we have included the word ‘directly’ in line 64. The reviewer is correct in the comment that the use of genetic data has been used for assessing varying aspect of primate dispersal, there are very few instances of direct observation being used to evaluate dispersal. Regarding the note on landscape genetics in primates, there are relatively few studies that have been published between 2003 (the inception of the field) and 2021 (the time the review article cited was published; Westphal et al 2021) using true landscape genetic methods. To our knowlegde, there has not been a signficant uptick in publish since the publish of the 2021 review.

Point 3: What kind of samples ...  were collected between 2015-2017 from 15 localities … ?

Response 3: Fecal samples was added at line 154.

Reviewer 2 Report

The authors present an interesting study on the spatial movement of lemurs. The manuscript is well-written and the statistical analyses are mostly appropriate. However, I have a few comments to be addressed.

The authors have decided to remove related individuals. There seems to be an ongoing discussion regarding this approach:

Waples, R. S. & Anderson, E. C. Purging putative siblings from population genetic data sets: A cautionary view. Mol. Ecol. 26(5), 1211–1224 (2017)

Peterman W, Brocato ER, Semlitsch RD, Eggert LS (2016) Reducing bias in population and landscape genetic inferences: the effects of sampling related individuals and multiple life stages. PeerJ, 4, e1813.

I think, this aspect could be better explained in the manuscript so that the audience can understand why you have run the analyses the way you did. Also, would any of the results be different with all individuals included? This would be interesting to know because the sample size in this study is comparatively low after the removal of related individuals (N = 62).  Since different approaches currently exist, it would be important to have this comparison in the manuscript as it would add to case studies exploring the impacts of related individuals in population – and landscape genetic analyses. This is also important because this decision reduces the number of individuals considerably in the current study, which is the basis for subsequent statistical analyses.

Minor comments:

Line 173: Change ‘then’ to ‘than’.

Lines 436 and 441: Delete one period at the end of the sentence (i.e., after ‘region’).

Lines 435/436 and 441/442: You refer to white circles here in the figure legends, but I am only able to see red circles. Please correct or clarify.  Also, I’m a bit confused about both figure legends (Fig. 8 and 9) because you refer to cool and warm colours. It remained unclear to me what you mean here. Please clarify.

Line 461: In the figure legend, the abbreviation ENS is not explained. Please add.

Line 678: Change ‘ogistical’ to ‘logistical’.

Lines 700, 775, 818, and 894: Species name should be italicized.

Lines 931/932: I think this citation is wrong. Have a look here: https://www.ecologyandsociety.org/vol4/iss1/art16/ and correct it accordingly.

Line 954: Add issue and page numbers.

Supplementary Material:

Lines 97, 113, 124: There is an unaccepted change track – please remove.

Author Response

Response to Reviewer 2 Comments

Point 1: The authors have decided to remove related individuals. There seems to be an ongoing discussion regarding this approach:

Waples, R. S. & Anderson, E. C. Purging putative siblings from population genetic data sets: A cautionary view. Mol. Ecol. 26(5), 1211–1224 (2017)

Peterman W, Brocato ER, Semlitsch RD, Eggert LS (2016) Reducing bias in population and landscape genetic inferences: the effects of sampling related individuals and multiple life stages. PeerJ, 4, e1813.

I think, this aspect could be better explained in the manuscript so that the audience can understand why you have run the analyses the way you did. Also, would any of the results be different with all individuals included? This would be interesting to know because the sample size in this study is comparatively low after the removal of related individuals (N = 62).  Since different approaches currently exist, it would be important to have this comparison in the manuscript as it would add to case studies exploring the impacts of related individuals in population – and landscape genetic analyses. This is also important because this decision reduces the number of individuals considerably in the current study, which is the basis for subsequent statistical analyses.

Response 1: Removal of relatives is common practice in population and landscape genetic assessments to avoid the introduction of bias and artifical genetic structuring that can arise when relatives are present in the sample, particularly in highly social group-living animals like primates. For example, see:

Rodrígues-Ramilo ST and Wang J. 2012. The effect of close relatives on unsupervised Bayesian clustering algorithms in population genetic structure analysis. Molecular Ecology Resources 12: 873-884.

We have added in the sentence “Removal of related individuals is generally recommended to avoid introducing bias and artificial structure in population and landscape genetic assessments” and the above citation at lines 160-162 to clarify this decision to the reader.

We appreciate Reviewer 2’s suggestion to retain siblings in our analyses (also noted in both papers above); however, based on our knowledge of the species (groups tend to cluster spatially into neighborhoods of parents and their dependent and subadult offspring; Baden et al. 2021) and our sampling regime (samples opportunistically collected from multiple individuals within the same social group), we chose to exclude relatives to reduce the likelihood of introducing artifical substructuring into our dataset, as per standard practice. While we agree with the reviewer that evaluating how including v.s. excluding related individuals might affect the results of landscape genetic analyses, this is beyond the scope of our current paper.

Point 2: Line 173: Change ‘then’ to ‘than’.

Response 2: This revision was made at line 160.

Point 3: Lines 436 and 441: Delete one period at the end of the sentence (i.e., after ‘region’).

Response 3: Periods were deleted at lines 419 and 429.

Point 4: Lines 435/436 and 441/442: You refer to white circles here in the figure legends, but I am only able to see red circles. Please correct or clarify.  Also, I’m a bit confused about both figure legends (Fig. 8 and 9) because you refer to cool and warm colours. It remained unclear to me what you mean here. Please clarify.

Response 4: We clarified that circles in the figure were red at lines 419 and 428. Additionally, we clarifed that areas of higher conductance were those of ‘darker greys’ and areas of hight resistance were those of ‘lighter greys’ in lines 416-417 and 426-427.

Point 5: Line 461: In the figure legend, the abbreviation ENS is not explained. Please add.

Response 5: We clarified the meaning behind ENS in line 444.

Point 6: Line 678: Change ‘ogistical’ to ‘logistical’.

Response 6: This spelling error was corrected in line 651.

Point 7: Lines 700, 775, 818, and 894: Species name should be italicized.

Response 7: Species names were italicized in lines 785, 791, 897, and 905.

Point 8: Lines 931/932: I think this citation is wrong. Have a look here: https://www.ecologyandsociety.org/vol4/iss1/art16/ and correct it accordingly.

Response 8: The citation was corrected in line 716.

Point 9: Line 954: Add issue and page numbers.

Response 9: The issue and page numbers were added to this citation in line 925.

Point 10: Supplementary Material:

Lines 97, 113, 124: There is an unaccepted change track – please remove.

Response 10: All outstanding track changes in the Supplementary Material have been removed.